# SOFA Score Plus Impedance Ratio Predicts Mortality in Critically Ill Patients Admitted to the Emergency Department: Retrospective Observational Study

**DOI:** 10.3390/healthcare10050810

**Published:** 2022-04-27

**Authors:** Ashuin Kammar-García, Lilia Castillo-Martínez, Javier Mancilla-Galindo, José Luis Villanueva-Juárez, Anayeli Pérez-Pérez, Héctor Isaac Rocha-González, Jesús Arrieta-Valencia, Miguel Remolina-Schlig, Thierry Hernández-Gilsoul

**Affiliations:** 1Dirección de Investigación, Instituto Nacional de Geriatría, Mexico City 10200, Mexico; kammar_nutrition@hotmail.com; 2Sección de Estudios de Posgrado e Investigación, Escuela Superior de Medicina, Instituto Politécnico Nacional, Mexico City 11340, Mexico; heisaac2013@hotmail.com (H.I.R.-G.); jearrval@yahoo.com.mx (J.A.-V.); 3Department of Clinical Nutrition, Instituto Nacional de Ciencias Médicas y Nutrición Salvador Zubirán, Mexico City 14080, Mexico; cam7125@gmail.com (L.C.-M.); viljjol@yahoo.com.mx (J.L.V.-J.); 4Facultad de Medicina, Universidad Nacional Autónoma de México, Mexico City 04360, Mexico; javimangal@gmail.com; 5Licenciatura en Nutrición, Facultad de Ciencias de la Salud, Universidad Autónoma de Tlaxcala, Tlaxcala 90750, Mexico; 6Emergency Department, Instituto Nacional de Ciencias Médicas y Nutrición Salvador Zubirán, Mexico City 14080, Mexico; anayeyip@gmail.com (A.P.-P.); mirems@yahoo.com (M.R.-S.)

**Keywords:** SOFA, impedance ratio, mortality, emergency department, critical care, prediction

## Abstract

Background: The Sequential Organ Failure Assessment (SOFA) is a scoring system used for the evaluation of disease severity and prognosis of critically ill patients. The impedance ratio (Imp-R) is a novel mortality predictor. Aims: This study aimed to evaluate the combination of the SOFA + Imp-R in the prediction of mortality in critically ill patients admitted to the Emergency Department (ED). Methods: A retrospective cohort study was performed in adult patients with acute illness admitted to the ED of a tertiary-care referral center. Baseline SOFA score and bioelectrical impedance analysis to obtain the Imp-R were performed within the first 24 h after admission to the ED. A Cox regression analysis was performed to evaluate the mortality risk of the initial SOFA score plus the Imp-R. Harrell’s C-statistic and decision curve analyses (DCA) were performed. Results: Out of 325 patients, 240 were included for analysis. Overall mortality was 31.3%. Only 21.3% of non-surviving patients died after hospital discharge, and 78.4% died during their hospital stay. Of the latter, 40.6% died in the ED. The SOFA and Imp-R values were higher in non-survivors and were significantly associated with mortality in all models. The combination of the SOFA + Imp-R significantly predicted 30-day mortality, in-hospital mortality, and ED mortality with an area under the curve (AUC) of 0.80 (95% CI: 74–0.86), 0.79 (95% CI: 0.74–0.86) and 0.75 (95% CI: 0.66–0.84), respectively. The DCA showed that combining the SOFA + Imp-R improved the prediction of mortality through the lower risk thresholds. Conclusions: The addition of the Imp-R to the baseline SOFA score on admission to the ED improves mortality prediction in severely acutely ill patients admitted to the ED.

## 1. Introduction

The Emergency Department (ED) is the first opportunity to generate therapeutic plans based on the severity and prognosis of disease of patients with acute illness. Different scoring systems have been designed to determine disease severity and to predict adverse outcomes in critically ill patients [1] and to improve the quality of therapeutic and preventive measures [2]. The Sequential Organ Failure Assessment (SOFA) is a scoring system used for the evaluation of disease severity and prognosis in critically ill patients [3]. It is based on the evaluation of six systems: respiratory, cardiovascular, neurological, hepatic, renal, and coagulation [4]. Although the SOFA score was not developed for the prediction of mortality, its usefulness to predict death has been observed in studies conducted in the Intensive Care Unit (ICU), demonstrating a close relationship between organ failure and mortality [5,6]. Recently, the use of the initial SOFA score has been validated as a good predictor of mortality [7]. Despite these findings, few studies have used the SOFA score for the prediction of mortality in non-ICU settings, such as the ED [3]. This scoring system has characteristics that make it suitable for the ED since it requires laboratory data often routinely measured upon ED admission [1]. Recent studies have suggested the inclusion of other mortality predictors, in addition to the SOFA score, could further improve the identification of high-risk patients [8], such as serum lactate levels [9], and C-reactive protein (CRP) [10].

On the other hand, fluid overload is an independent factor associated with a worse prognosis in critically ill patients [11], which prolongs multiorgan dysfunction [12]. New markers of fluid overload such as the impedance ratio (Imp-R) have been associated with a worse prognosis in critically ill patients [13,14]. The imp-R is the ratio between high- and low-frequency impedance values (200/5 kHz) obtained during bioelectrical impedance analysis (BIA), which contemplates total body water (impedance at 200 kHz) and extracellular water (impedance at 5 kHz). It indicates conduction in these body fluid compartments. The penetration of current into cells is frequency-dependent, thus, the 200/5 kHz index indicates the ratio of higher-to-lower current that enter the cells. If the difference between these two values becomes smaller over time, it may indicate that the cell is becoming less healthy. The resistance of the cell membrane at 5 kHz is significantly reduced in the case of critical illness, and the difference between the impedance values at 5 and 200 kHz is markedly closer to each other, indicating cellular deterioration [13,14]. Imp-R values of ≤0.78 in males and ≤0.82 in females have been observed in healthy individuals and values approaching 1.0 suggest that the two measured impedances are approaching each other in value [15]. The Imp-R has been previously evaluated as a predictor of mortality in critically ill patients [13,14,15,16].

Since multiorgan dysfunction and fluid overload are conditions associated with mortality in critically ill patients, their combination could possibly improve mortality prediction in patients admitted to the ED who develop critical disease. Therefore, the aim of this study was to evaluate the combination of the initial SOFA score and the Imp-R to predict mortality in critically ill patients admitted to the ED.

## 2. Methods

This was a retrospective observational study performed in a cohort of patients [17] admitted to the ED of the Instituto Nacional de Ciencias Médicas y Nutrición Salvador Zubirán—a tertiary care referral center in Mexico City—between September 2016 and September 2019. Adult (≥18 years) patients with acute illness (defined as any disease that develops quickly, is intense or severe, and generally lasts a relatively short period of time, often less than 1 month) who were admitted to a hospital bed in the Emergency Department, within the 24 h prior to assessment for bioelectrical impedance measurement were eligible for inclusion in the study. Patients without acute illness who were not severely ill were not eligible to be included in the study. Patients who had metal prostheses or who had errors on bioelectrical impedance measurements were excluded from the study. Patients without all necessary clinical and laboratory variables to calculate the SOFA score within the first 24 h after admission, or who were lost to follow-up were eliminated. The study protocol was approved by the Ethics Committee of the Instituto Nacional de Ciencias Médicas y Nutrición Salvador Zubirán under number 1977.

### 2.1. Data Collection and Management

All clinical (BMI, vasopressor assistance, mechanic ventilation, Glasgow Coma Scale) and biochemical (blood urea nitrogen, creatinine, sodium, potassium, C-reactive protein, platelet count, bilirubin, albumin, lactate) variables, as well as the cause of hospitalization and comorbidities were obtained directly from electronic medical records. Hospital stay was calculated from the first day of admission to the ED until the day of hospital discharge.

The initial SOFA score (range 0 to 24) was calculated and evaluated for each patient by using the first value of the physiological (partial pressure of oxygen, PaO_2_; fraction of inspired oxygen, FiO_2_; mean arterial blood pressure, MAP; Glasgow Coma Scale, and urine output) and laboratory (platelet count, bilirubin, and creatinine) parameters obtained within the first 24 h after admission to the ED [7].

At the time of admission, all patients underwent a bioelectrical impedance measurement once. Patients were placed in a supine position and any metal objects in contact with the body were removed; arms and legs were abducted and, in order to avoid contact between the thighs, obese patients had a sheet placed between their legs to avoid their contact. Two electrodes were placed dorsally on one hand (at the third metacarpophalangeal and carpal joint) and two other electrodes on one foot (at the third metatarsophalangeal joint and tarsal joint).

Using a tetrapolar, portable (length 240 mm, width 155 mm, and height 30 mm) impedance analyzer (BODYSTAT QuadScan 4000; BOSYSTAT LTD, Isle of Man, UK) with an alternating current of 800 mA at four different frequencies (5, 50, 100, and 200 kHz), the impedance values (Z) of all frequencies were obtained and transformed into resistance (R) and reactance (Xc) at 50 kHz. The impedance ratio (Imp-R) was calculated as the quotient of Z at 200 kHz between Z at 5 kHz calculated to reflect the total body water and extracellular water compartment, respectively [16]. Two clinicians standardized and well-trained on the tetrapolar method performed BIA measurements [18]. The time required for BIA measurement was approximately 5 min for each person. The calibration of the equipment was checked periodically by a test resistor with a known value of 500 U (range 496–503 Ω) [19]. 

All patients were followed up for 30 days from their ED admission. The incidence of mortality was obtained directly from hospital records or through telephone interviews with family members. The primary endpoints of this study were 30-day all-cause mortality, in-hospital mortality, and ED mortality.

### 2.2. Sample Size Calculation

The sample size was calculated according to an estimate of mortality risk according to increases in the SOFA score; previous studies showed a nine-fold higher risk for patients with an increase in one or more points, on admission to the ICU from the ED [20]. Being more conservative with the increase in risk, we considered an estimate of a two-times higher risk for every one-point increase in the SOFA score and considered a mortality of 15% in critical patients [20], which yielded a minimum sample size of 220 patients with an alpha error of 0.05 and power of 80%.

### 2.3. Statistical Analyses

All descriptive data are summarized as the median with the inter quartile range (IQR = 1st–3rd quartile) or as a frequency with a percentage. Comparisons between the groups were performed using a Mann-Whitney U test. 

Different Cox regression models were applied to estimate the 30-day mortality risk, in-hospital mortality, and ED mortality, according to the initial SOFA score or Imp-R. Variables were entered into the models as continuous quantitative variables. The models were adjusted for: age, sex, BMI, invasive mechanical ventilation, creatinine, and lactate. Results of all models are summarized as hazard ratios with 95% confidence intervals (95% CI). Furthermore, models were plotted in cubic splines.

Other models were created in which the SOFA score and the Imp-R were included in the same model to determine if both variables could predict mortality better. The univariate and combined models were compared using Akaike Information Criteria (AIC) and Bayesian Information Criteria (BIC). The predictive value of each model was calculated by Harrell’s C-statistic and expressed with the area under the curve (AUC). The evaluation and comparison of the baseline SOFA score model and its combination with Imp-R was carried out through a decision curve analysis.

The model assumptions were verified by residual analysis. All statistical analyses and figures were carried out in the statistical software SPSS version 21 and R v.3.6.1. A *p* < 0.05 was considered as statistically significant.

## 3. Results

The flow of patients is shown in Figure 1. Out of 325 patients, 240 were included for analysis. Demographic and clinical characteristics of patients at the ED admission are summarized in Table 1. Most patients admitted to the ED were women (58.3%) over 60 years (47.9%), with a median age of 60 (46–71.8) years. The main causes of admission to the ED were gastrointestinal (30%), infectious (21.7%), and cardiovascular (15.4%). The presence of kidney disease (16.3%) and cirrhosis (16.7%) were similarly frequent. Most patients (70.8%) did not require vasopressor support during their hospital stay, whereas 12.9% of patients required mechanical ventilation with an approximate duration of 2 (1–5) days. The median hospital stay was 6 (2–12) days. The incidence of mortality was 31.3% (n = 75), with most deaths occurring in hospital (78.4%, n = 59) rather than after discharge (21.3%, n = 16). Of all the in-hospital deaths, 40.6% (n = 24) died during their stay in the ED. 

Regarding the SOFA scores, 58.3% of patients admitted to the ED had an initial SOFA score of 2 to 7, with a median score of 6 (IQR: 4–9). In non-surviving patients, the initial SOFA score was higher (9, IQR: 6–11) than for survivors (5, IQR: 3–7; *p* < 0.001). A similar situation was observed in patients who died in hospital (5, IQR: 3–7.5 vs. 9, IQR: 6–12; *p* < 0.0001) or in the ED (6, IQR: 4–9 vs. 9, IQR: 6–11.7; *p* = 0.001; Table 2).

The Imp-R on admission to the ED was 0.85 (IQR: 0.81–0.88). When the values of the Imp-R were compared between survivors and non-survivors, higher values were observed in non-survivors in the different mortality groups: 30-day mortality (0.84, IQR: 0.80–0.87 vs. 0.87, IQR: 0.83–0.90; *p* < 0.001), in-hospital mortality (0.84, IQR: 0.80–0.87 vs. 0.87, IQR: 0.83–0.90; *p* < 0.001), and ED mortality (0.84, IQR: 0.80–0.88 vs. 0.88, IQR: 0.85–0.90; *p* < 0.001) (Table 2). Of the other BIA parameters, only reactance and phase angle showed differences between survivors and non-survivors in all mortality groups (Appendix A).

Table 3 shows the results of the Cox regression analyses for the different mortality models. Increasing values of the initial SOFA score and the Imp-R were associated with a higher mortality risk. Each additional point in the initial SOFA score increased the 30-day mortality, in-hospital mortality, and ED mortality risks by 11%, 21%, and 18%, respectively. Likewise, each 0.01 unit increase in the Imp-R led to larger 30-day mortality, in-hospital mortality, and ED mortality risks by 9%, 9%, and 12%, respectively. Figure 2 shows the splines of each mortality model according to the initial SOFA score and the Imp-R; mortality risk begins to increase significantly at values higher than five points for the initial SOFA score and 0.85 for the Imp-R.

The AUCs of the initial SOFA score were 0.74 (95% CI: 0.68–0.81), 0.77 (95% CI: 0.70–0.83), and 0.71 (95% CI: 0.61–0.81) for the 30-day mortality, in-hospital and ED mortality, respectively (all *p* < 0.001). Conversely, the AUC of the combination of initial SOFA score plus Imp-R were 0.80 (95% CI: 0.74–0.86), 0.80 (95% CI: 0.74–0.86), and 0.75 (95% CI:0.66–0.84), respectively (all *p* < 0.001). The comparison of models for the prediction of mortality by information criteria showed that, for each mortality model, the combination of the initial SOFA score with the Imp-R improved the outcome prediction (30-day mortality: ΔAIC = 11.46, ΔBIC = 9.14, in-hospital mortality: ΔAIC = 4.73, ΔBIC = 2.65, and ED mortality: ΔAIC = 4.81, ΔBIC = 3.64).

The decision curves for the different mortality models are shown in Figure 3. For the initial SOFA score and the combination with the Imp-R, a slight superiority of the latter can be observed since it improves the prediction of mortality at the lower risk thresholds.

## 4. Discussion

In this study, the combination of the initial SOFA score and the Imp-R at admission to the ED was accurate at predicting mortality in patients with acute illness admitted to the ED of a tertiary care referral center in Mexico City. The combination of SOFA plus the Imp-R showed a better prediction of 30-day mortality, in-hospital mortality, and ED mortality than the initial SOFA score alone.

To our knowledge this is the first study addressing the use of a marker of fluid overload in combination with the SOFA score. Our aim was to assess if the evaluation of fluid overload could aid the prediction of mortality by the SOFA score. To this end, we used the Imp-R as a marker of fluid overload, which has already been considered as a prognostic marker of mortality [14] and as a marker of fluid overload [16]. The Imp-R and the SOFA score were individual predictors of mortality in all of the models evaluated (30-day mortality, in-hospital mortality, and ED mortality), and this is similar to what has been reported previously [13].

In previous studies, predictors of mortality in critically ill patients have been characterized, which include different biological parameters such as creatinine, lactate, bilirubin, and CRP, as well as clinical parameters such as heart rate, blood pressure, hospital stay, mechanical ventilation, and fluid overload [4,8,16,21]. Each of these parameters has a pathophysiological role in organ deterioration of the critically ill patient, although they do not allow the determination of multiorgan compromise by themselves; consequently, the SOFA score takes a leading role in the evaluation of the prognosis of patients with severe acute illness. The SOFA score includes predictors of mortality such as creatinine, bilirubin, and mechanical ventilation, as well as other parameters like blood pressure, instead of the heart rate or lactate. Some studies have shown that changes in the SOFA score can improve the prediction of mortality [4,22,23]. These changes are mainly substitutions of some parameters for others, which are easier to obtain during a hospital stay. 

The SOFA score has been tested in multiple different contexts to predict mortality [4,5,20,23]. For instance, by applying it at different times since admission to the ICU or by calculating average scores during a certain period of time. Recently, attempts to combine the SOFA score with other mortality risk factors have been carried out. For example, by combining it with CRP [24], procalcitonin [25], or lactate [26].

Studies that evaluated the prediction of hospital mortality through the SOFA score at ICU admission showed an AUC of between 0.63 and 0.82 [1,4], whereas studies that evaluated the SOFA score after admission to the ICU (40–72 h) have shown a higher AUC: 0.85–0.95 [5,18]. Nonetheless, these estimates may be biased since there is a risk that interventions performed after 24 h could affect mortality risk and, therefore, the scores too [7], thereby altering their predictive values. The results of this study are in line with the findings of others aiming at predicting in-hospital mortality [1], although our study has an additional strength since we evaluated 30-day mortality, in-hospital mortality, and mortality in the ED, simultaneously. 

Other studies have shown that combining the SOFA score with other mortality markers increases the mortality prediction. With the combination of the SOFA with procalcitonin (PCT), the single SOFA score had an AUC of 0.86, while its combination with PCT showed an AUC of 0.91 [25]. In the combination of the SOFA score with lactate, the AUC was 0.83, which was higher than the SOFA alone and other prognostic scales [26]. The N-terminal pro-brain natriuretic (NT-proBNP) has also been used in combination with the SOFA score, being a stronger predictor of hospital mortality than either variable alone [27]. 

The combination of the SOFA + the Imp-R are mutually complementary since there is a relationship between multiorgan dysfunction, increased adverse events, and fluid overload [12]. Other combinations of the SOFA score with markers such as lactate may present discrepancies, since the increase in lactate in critically ill patients can indicate hypoperfusion, something which is already considered by the SOFA score with the mean arterial pressure, which is why adding lactate to the prediction could only modestly improve mortality prediction, since patients with the lowest MAP could be the same with the highest lactate levels. Likewise, CRP has been shown in other studies to be an important marker of a patient’s prognosis [28], although its combination with the SOFA does not improve the mortality prediction in critically ill patients [23].

All studies that have evaluated combinations of the SOFA scores with other mortality markers have been performed in patients admitted to the ICU, an approach which is different to ours since we performed evaluations of patients upon admission to the ED. This is relevant, since treatments throughout the hospital stay could affect the scores and mask the true initial differences [7]. Thus, our findings could be generalizable to patients with acute illness who are admitted to the ED and who have these measurements performed upon admission. For these same reasons, our findings may not be generalized to patients who develop severe illness later during their hospital stay. 

The use of the initial SOFA score in combination with the Imp-R for prognostic purposes is non-invasive, which could suggest that using these tools as predictors of mortality in the ED could be viable. Furthermore, determining fluid overload upon admission could be relevant to guide management since the abuse of resuscitation fluids, due to poor evaluation of organ function and fluid volume status, could lead to the deterioration of severely ill patients [29,30]. Our results contribute to continuing to improve common and widely used mortality prediction models such as the SOFA score.

The main limitations of this study are its retrospective observational design, the fact that it was performed in a single tertiary care referral center, and the lack of repeated measures of the Imp-R to elucidate how it may change in survivors. Another limitation was the relatively small sample size because only patients for whom all laboratory results or clinical parameters were available for the calculation of the SOFA score could be included in the study. This could have left patients admitted with diseases of lower severity out of the study, since exhaustive laboratory determinations are not regularly performed for less severely ill patients. At the same time, this limits the clinical applicability of our findings since the array of patients for whom these calculations could be readily performed in the ED may be low. Furthermore, we were unable to assess the time from onset of the illness to the ED admission, which could be a relevant confounder since patients who received delayed medical care could have a higher mortality risk [31]. Similarly, a limitation of this study is the use of a single severity scale and not considering other scales that have been validated to be used in critically ill patients (APACHE, SAS, or MEXSOFA) or other common evaluation tools used in the ED (qSOFA, NEWS, or MEDS). It would be interesting to carry out more research in to the combination of these scales with the Imp-R to determine if there is an improvement in the prediction of mortality for patients admitted to the ED, mainly in the qSOFA scale that has become widely used, since it is simpler to calculate [32]. Some studies have already begun to explore the combinations of the qSOFA with biomarkers in ICU patients [33,34,35] but more research is needed to better determine their use in the ED. 

Evaluating the Imp-R prospectively in patients admitted to the Emergency Department without critical disease could allow the determination of reference values to compare critical patients against. We currently know that normal values for the Imp-R in the healthy patients are ≤0.78 in males and ≤0.82 in females [36], but reference values have not been determined for less severely ill patients in the ED.

## 5. Conclusions

The initial SOFA score and the Imp-R upon admission to the ED are independent predictors of 30-day mortality, in-hospital mortality, and ED mortality. The addition of the Imp-R to the baseline SOFA score on admission to the ED improves mortality prediction in severely acutely ill patients. This new assessment strategy could provide additional information to inform the prognosis of patients admitted to the ED with severe acute illness in the future, although its current clinical applicability may be limited due to low availability.

## Figures and Tables

**Figure 1 healthcare-10-00810-f001:**
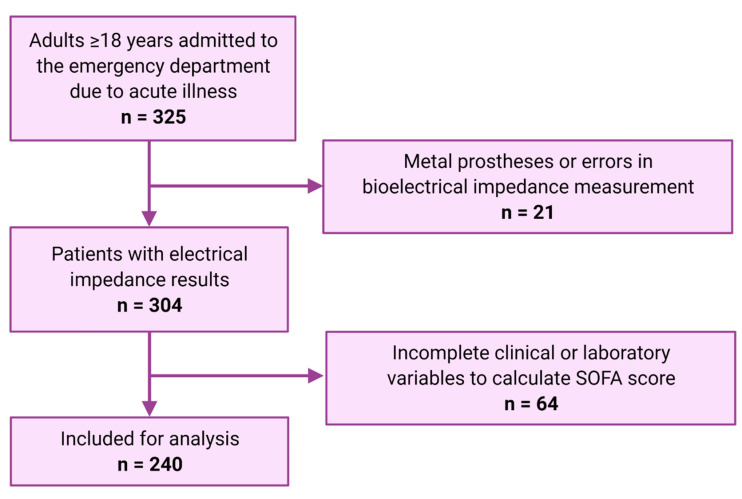
Flow of patients assessed for eligibility.

**Figure 2 healthcare-10-00810-f002:**
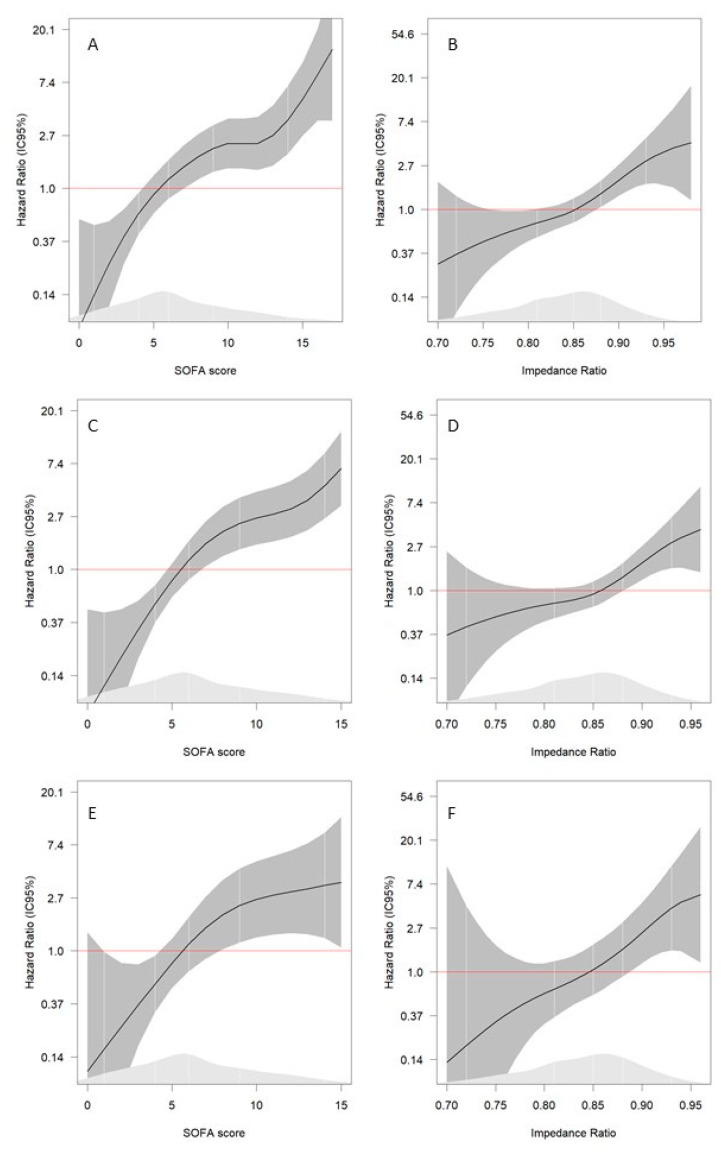
Splines of initial SOFA score and impedance ratio for prediction of mortality. (**A**) Prediction of 30-day mortality by initial SOFA score. (**B**) Prediction of 30-day mortality by impedance ratio. (**C**) Prediction of in-hospital mortality by initial SOFA score. (**D**) Prediction of in-hospital mortality by impedance ratio. (**E**) Prediction of ED mortality by initial SOFA score. (**F**) Prediction of ED mortality by impedance ratio. 30-day mortality model adjusted by: age, sex, and body mass index, invasive mechanic ventilation, creatine, lactate. In-hospital mortality model adjusted by: age, sex, and body mass index, invasive mechanic ventilation. ED mortality model adjusted by: age, sex.

**Figure 3 healthcare-10-00810-f003:**
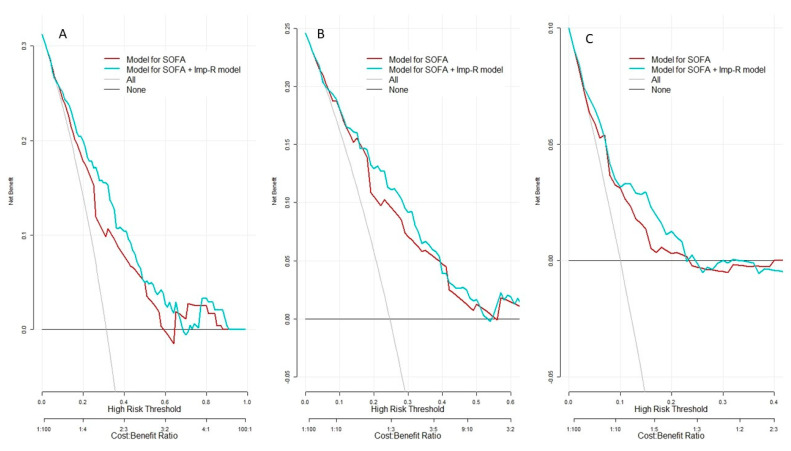
Decision curve analyses for initial SOFA score model and the combination with impedance ratio in the prediction of mortality. (**A**) Models for prediction of 30-day mortality, (**B**) Models for prediction of in-hospital mortality, (**C**) Models for prediction of ED mortality.

**Table 1 healthcare-10-00810-t001:** Demographic and clinical characteristic at ED admission.

Variables	Total Sample n = 240
Sex, n (%)	
Female	140 (58.3)
Male	100 (41.7)
Age, years	60 (46–71.8)
BMI, kg/m^2^	24.5 (21.5–28.3)
<18	21 (8.8)
18–24.9	113 (47.1)
25–29.9	70 (29.2)
30–34.9	21 (8.8)
35–39.9	8 (3.3)
≥40	7 (2.9)
Causes of hospitalization, n (%)	
Neurology	12 (5)
Cardiovascular	37 (15.4)
Respiratory	23 (9.6)
Gastrointestinal	72 (30)
Oncology	7 (2.9)
Endocrinology	9 (3.8)
Nephrology	19 (7.9)
Rheumatology	2 (0.8)
Infection	54 (21.7)
Hematology	7 (2.9)
Comorbidity, n (%)	
Diabetes	74 (30.58)
Hypertension	74 (30.8)
Renal failure	39 (16.3)
Hepatic cirrhosis	40 (16.7)
Malignancy	49 (20.4)
VIH	8 (3.3)
Use of Vasopressors, n (%)	70 (29.2)
Use of a mechanical ventilator, n (%)	
Yes	31 (12.9)
No	209 (87.1)
Initial SOFA score, n (%)	6 (4–9)
0–1	17 (7.1)
2–7	140 (58.3)
8–11	56 (23.3)
>11	27 (11.3)
30-days mortality, n (%)	75 (31.3)
In-hospital mortality, n (%)	59 (24.6)
ED mortality, n (%)	24 (10)

Data are expressed by median and IQR (1st–3rd quartile) or frequency and percentage (%); BMI: Body mass index, SOFA: Sequential Organ Failure Assessment, ED: Emergency Department.

**Table 2 healthcare-10-00810-t002:** Comparison of clinical data, bioimpedance analyses, and biochemical analyses at admission to ED in survivors and non-survivors.

	30–Days Mortality			In–Hospital Mortality			In–ED Mortality		
	No n = 165	Yes n = 75	*p* Value	No n = 181	Yes n = 59	*p* Value	No n = 216	Yes n = 24	*p* Value
Age, years	57 (40.5–67)	64 (51–76)	0.005	58 (42.5–69)	64 (50–76)	0.036	59 (44.3–69)	73 (54–80.1)	0.009
BMI	24.6 (21.7–28.1)	24.1 (20.8–24.1)	0.735	24.6 (21.6–28.1)	24 (20.9–28.7)	0.998	24.2 (21.4–27.7)	26.9 (21.9–30)	0.159
Initial SOFA score	5 (3–7)	9 (6–11)	<0.001	5 (3–7.5)	9 (6–12)	<0.001	6 (4–9)	9 (6–11.7)	0.001
Impedance ratio	0.84 (0.8–0.87)	0.87 (0.83–0.9)	<0.001	0.84 (0.8–0.87)	0.87 (0.83–0.9)	<0.001	0.84 (0.8–0.88)	0.88 (0.85–0.9)	0.002
Creatinine, mg/dL	1.1 (0.72–2)	1.6 (0.81–2.68)	0.109	1.1 (0.73–2.08)	1.62 (0.87–2.75)	0.098	1.21 (0.75–2.1)	2.06 (1–4.3)	0.077
CRP, mg/L	5.2 (0.84–15.9)	10.7 (5.5–15.3)	0.056	5 (1.1–15.7)	11.9 (6.7–15.5)	0.028	7.1 (1.6–15.5)	14.1 (11.3–21.9)	0.017
Bilirubin, mg/dL	0.69 (0.47–1.5)	1.66 (0.64–6.7)	<0.001	0.7 (0.48–1.51)	1.95 (0.64–8.67)	<0.001	0.76 (0.5–1.81)	3.39 (0.82–6.6)	0.004
Lactate, mg/dL	1.7 (1.2–2.8)	3.4 (1.9–6.2)	<0.001	1.8 (1.3–2.8)	4.1 (2.2–6.7)	<0.001	2 (1.4–3.8)	3.3 (2–6.9)	0.005

Data are expressed by median and IQR (1st–3rd quartile); ED: Emergency Department, BMI: Body mass index, SOFA: Sequential Organ Failure Assessment, Z: impedance, CRP: C-reactive protein.

**Table 3 healthcare-10-00810-t003:** Unadjusted and adjusted Cox regression models for prediction of mortality by initial SOFA score and impedance ratio.

	Unadjusted Model			Adjusted Model		
	β Coefficient	HR (95% CI)	*p* Value	β Coefficient	HR (95% CI)	*p* Value
30-days mortality model ^a^						
Initial SOFA score	0.16	1.18 (1.10–1.27)	<0.001	0.11	1.12 (1.03–1.22)	0.012
Impedance ratio	0.10	1.11 (1.05–1.17)	<0.001	0.09	1.10 (1.04–1.16)	0.002
In-hospitality mortality model ^b^						
Initial SOFA score	0.22	1.25 (1.16–1.34)	<0.001	0.21	1.23 (1.14–1.33)	<0.001
Impedance ratio	0.10	1.10 (1.05–1.17)	<0.001	0.09	1.10 (1.03–1.16)	0.002
ED mortality model ^c^						
Initial SOFA score	0.19	1.20 (1.09–1.34)	<0.001	0.18	1.20 (1.08–1.33)	0.001
Impedance ratio	0.14	1.15 (1.05–1.25)	0.002	0.12	1.13 (1.03–1.24)	0.014

Adjusted model for: a: age, sex, and body mass index, invasive mechanic ventilation, creatine, lactate; b: age, sex, and body mass index, invasive mechanic ventilation; c: age, sex. SOFA: Sequential Organ Failure Assessment, ED: Emergency Department.

## Data Availability

The data presented in this study are available on request from the corresponding author. The data are not publicly available due to institutional privacy protection policies.

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
