# Peer review of "SOFA Score Plus Impedance Ratio Predicts Mortality in Critically Ill Patients Admitted to the Emergency Department: Retrospective Observational Study"

_healthcare, 2022, doi:10.3390/healthcare10050810_

Round 1

Reviewer 1 Report

Dear authors;

I appreciate your effort for this valuable study. 

Please find my comments below;

- The number of patients are appropriate 

- The title is in harmony with the content

- The references are current and the number of them are enough

- The discussion section is satisfactory

- Limitations and conclusion sections are appropriate

- My decision is accept

Kind regards.... 

Author Response

April 21, 2022

Dr. Rahman Shiri

Editor-in-Chief

Healthcare

Thank you and the reviewers for kindly reviewing our paper entitled SOFA score plus impedance ratio predict mortality in critically ill patients admitted to the emergency department: Retrospective observational study. Manuscript ID: healthcare-1673018. We appreciate the thoughtful comments and suggestions which have allowed us to improve our manuscript. We have prepared detailed responses to all of your comments below.

Reviewer 1:

Dear authors;

I appreciate your effort for this valuable study. 

Please find my comments below;

- The number of patients are appropriate 

- The title is in harmony with the content

- The references are current and the number of them are enough

- The discussion section is satisfactory

- Limitations and conclusion sections are appropriate

- My decision is accept

R: Thank you very much for the time spent in reviewing our manuscript and for your positive comments.

Reviewer 2 Report

Overall well written and designed study, despite the retrospective nature of the study design.  
Interesting use of the Imp-R for initial assessment of sick patients, but further details about the use in the Imp-R measurements  is necessary.  I also feel some of the tables need to be tidied up, which I have commented upon below. 
my other critique is the lack of controls.  I know this is retrospective, but impedance measurements could be taken on random ED patients who aren't acutely ill, to provide some context as to the range/values normally seen with the impedance testing.  

Methods:
-line 75 - define "acute illness".  Someone presenting with a heart attack would have an acute illness, but wouldn't get a SOFA score

-need much more information about the impedance testing - who does it? how long does it take? one measurements or 10 that are averaged? what is it measuring? done by everyone or only specialists? small machine or large device in one location?  portable device that an ED physician can use?  Take a lot of training to use this?  Expensive device?  Etc etc....your whole paper discusses this as a measurement to use in sick patients, but no information about this measurements are given.

Results:
for all results - please tell us what values are in the brackets?  Is this the confidence interval?  Mean values followed by the CI?  Please better define what values you are providing. (I see this is done in respect to the AUC discussion, but needs to be throughout)

line 137 - BMI - doesn't need to be here.  this is in the table.  you say most patients had normal BMI, but then say this was 47.1% - this is not most patients.  

any ability to determine whether patients did not receive vasopressors or intubation because their goals of care did not allow those interventions?

what is the Imp-R in normal patients?  what about patients not in the ED?  what about in ED patients in for other reasons?  There is a perfect group of controls there, but that wasn't tested - missed opportunity.  Should have been able to age/gender match for other ED patients who aren't acutely ill, and see what their Imp-R is.

what do these Imp-R values mean?  The range is very small, and on quick glance 0.84 and 0.87 aren't very different, especially with the CI provided (is this the CI?  see earlier comment about values provided)

Table 2:
What do all the impedance results mean?  Perhaps only have the sofa score, impedance ratio and lab values in this table?  I feel all the other impedance values only confuse the table and the reader.  remove, put in the text or include as suppl information

where any impedance values repeated?  Would have been interesting to see in the survivors the impedance values improved, where in those who died the impedance values worsened.

Table 3 - confused as to what this is showing.  The text (line 167 onwards) comments about how increasing values of initial SOFA and Imp-R were associated with higher mortality risk, but in the table it just says "initial sofa score" "initial impedance ratio"

Discussion:
while I applaud the authors when they comment that the SOFA score is not an ED based scoring system, but that they tried to use in on this population.  I wonder about applicability of the SOFA score in the ED (other than a research instrument), as it has many components, can be complicated, requires lab values and doesn't have meaningful impact to ED patients.  The authors try to show that doing it in the ED with the use of the Imp-R testing may have some usefulness in the ED, but I question the generalization of its use.  

was the parameters of the qSOFA score assessed as well?  This would be an interesting addition to this manuscript, as the qSOFA score is a fast measure, that doesn't require lab values, which might be more widely used in the EDs (although recent sepsis guidelines recommend not using as a screening tool) - I do see the qSOFA was commented upon in the discussion.

Author Response

April 21, 2022

Dr. Rahman Shiri

Editor-in-Chief

Healthcare

Thank you and the reviewers for kindly reviewing our paper entitled SOFA score plus impedance ratio predict mortality in critically ill patients admitted to the emergency department: Retrospective observational study. Manuscript ID: healthcare-1673018. We appreciate the thoughtful comments and suggestions which have allowed us to improve our manuscript. We have prepared detailed responses to all of your comments below.

Reviewer 2:

Overall well written and designed study, despite the retrospective nature of the study design.  
Interesting use of the Imp-R for initial assessment of sick patients, but further details about the use in the Imp-R measurements is necessary.  I also feel some of the tables need to be tidied up, which I have commented upon below. 

R= Thank you very much for your comments. We have added new information in the introduction which we believe will help to better understand how the Imp-R is calculated and applied. Furthermore, we have made the solicited changes to the tables.

My other critique is the lack of controls.  I know this is retrospective, but impedance measurements could be taken on random ED patients who aren't acutely ill, to provide some context as to the range/values normally seen with the impedance testing.

R: Thank you for observing this and commenting on it. We agree that having healthy controls would be very informative to understand how our results compare versus healthy patients. Nonetheless, since the main objective of our observational study is prognostic, which requires obtaining measurements and collecting data on outcomes of patients throughout their hospital stay, including healthy controls would be technically difficult and potentially unethical since we would have to hospitalize healthy patients to observe them while admitted to hospital. We believe that a lack of explanations on our behalf could have led to this misunderstanding since it is entirely logic that impedance measurements can be performed on random patients evaluated in the ED. However, our study included only patients who were admitted and assigned a bed in the ED, with further in-hospital follow-up. Thus, we have carefully detailed better the setting of our study and its participants. Despite the lack of comparison with healthy controls, we believe that our study was adequate for our purpose of using this as a prognostic tool in patients who are admitted to hospital and assigned a bed in the ICU to determine prognosis.

Methods:
-line 75 - define "acute illness".  Someone presenting with a heart attack would have an acute illness, but wouldn't get a SOFA score

R: Thank you, we have added the following definition for acute illness: “patients with acute illness (defined as any disease that develops quickly, is intense or severe, and generally lasts a relatively short period of time, often less than 1 month)”. We have added this definition in the manuscript. To further complement this definition, the MeSH term for Intensive Care Unit can be informative since it defines an ICU as: “Hospital units providing continuous surveillance and care to acutely ill patients”. Thus, acutely ill is commonly used to refer to patients who require intensive care. We chose to use the word “acutely ill” instead of “critically ill” to avoid confusions since the setting of our study was patients admitted to hospital beds in the emergency department who needed intensive care but were not able to be immediately admitted to the ICU due to the severity of their disease which does not allow in-hospital transportation, as well as hospital saturation in our center. We have also added the lack of representativeness of less severely ill patients to the limitations of the study.

-Need much more information about the impedance testing - who does it? how long does it take? one measurements or 10 that are averaged? what is it measuring? done by everyone or only specialists? small machine or large device in one location?  portable device that an ED physician can use?  Take a lot of training to use this?  Expensive device?  Etc etc....your whole paper discusses this as a measurement to use in sick patients, but no information about this measurements are given.

R: Thank you for the suggestion, we have added this information in the methods section.

 Results:
for all results - please tell us what values are in the brackets?  Is this the confidence interval?  Mean values followed by the CI?  Please better define what values you are providing. (I see this is done in respect to the AUC discussion, but needs to be throughout)

R: Thank you for the observation. The values in the brackets are the interquartile range (IQR), corresponding to the first and third quartile or what is the same, 25th and 75th percentiles. We have included the definition of IQR in the statistical analysis section as well as its acronym on all pertaining results.

line 137 - BMI - doesn't need to be here.  this is in the table.  you say most patients had normal BMI, but then say this was 47.1% - this is not most patients. 

R: Thank you for the observation. We have eliminated this phrase and added the BMI classification to table 1.

any ability to determine whether patients did not receive vasopressors or intubation because their goals of care did not allow those interventions?

R: Thank you for your question. All patients included in the study received medical attention according to international guidelines. No patients with no-resuscitation orders were included in this study.

what is the Imp-R in normal patients?  what about patients not in the ED?  what about in ED patients in for other reasons?  There is a perfect group of controls there, but that wasn't tested - missed opportunity.  Should have been able to age/gender match for other ED patients who aren't acutely ill, and see what their Imp-R is.

R: Thank you for your comment. As we have previously commented on one of our previous responses, it was not feasible to include such healthy controls. Regarding normal values for Imp-R, these are ≤0.78 in males and ≤0.82 in females (doi:10.1186/s12882-019-1511-y). We have added this in the methods section. We have also commented on the lack of healthy controls as a potential limitation of the study.

What do these Imp-R values mean?  The range is very small, and on quick glance 0.84 and 0.87 aren't very different, especially with the CI provided (is this the CI?  see earlier comment about values provided)

R: thank you for your question. Imp-R values reflect the total body water (impedance at 200 kHz) and extracellular water (impedance at 5 kHz). If the difference between these two values becomes smaller over time, this may indicate that the cell is becoming less healthy. The resistance of the cell membrane at 5 kHz is significantly reduced in the case of critical illness, and the difference between the impedance values at 5 and 200 kHz is markedly closer to each other, indicating cellular deterioration.

A paragraph explaining the meaning of Imp-R values has now been included in the introduction.

Table 2: What do all the impedance results mean?  Perhaps only have the sofa score, impedance ratio and lab values in this table?  I feel all the other impedance values only confuse the table and the reader.  Remove, put in the text or include as suppl information

R: Thank you for the suggestion. We agree that these values could confuse readers and agreed to move them to supplementary information.

Where any impedance values repeated?  Would have been interesting to see in the survivors the impedance values improved, where in those who died the impedance values worsened.

R: We totally agree that this would have been a very interesting and important thing to do. Unfortunately, our center only counts with one bioimpedance analyzer which belongs to the Emergency Department, reason why it was not feasible to perform subsequent determinations on other hospital floors or units. We have added this as a limitation of our study.

Table 3 - confused as to what this is showing.  The text (line 167 onwards) comments about how increasing values of initial SOFA and Imp-R were associated with higher mortality risk, but in the table it just says "initial sofa score" "initial impedance ratio"

R: Thank you for your comment. Table 3 shows the results of Cox regression analyses for every mortality model, considering initial SOFA score as a quantitative variable, as well as for Imp-R. What we declare in the manuscript aims is based on the results of regression analyses where every regression coefficient (β coefficient) shows a positive value, indicating that for every increasing unit of the SOFA score of the Imp-R, the risk of the outcomes being evaluated increase. This is supported by what is being presented in figure 2, where one can visually appreciate that the HR increases as the units of initial SOFA score and Imp-R also increase.  

Discussion:
while I applaud the authors when they comment that the SOFA score is not an ED based scoring system, but that they tried to use in on this population.  I wonder about applicability of the SOFA score in the ED (other than a research instrument), as it has many components, can be complicated, requires lab values and doesn't have meaningful impact to ED patients. The authors try to show that doing it in the ED with the use of the Imp-R testing may have some usefulness in the ED, but I question the generalization of its use. 

R:  Thank you very much for this insightful comment with which we agree since determining SOFA score in the ED alongside Imp-R can be challenging and will most likely not be widely available. Thus, its clinical application could be limited at present time. This could change as laboratory determinations and impedance analyzers become more accessible. Despite these limitations, we believe that our study provides interesting insights like showing that widely used prediction scores such as SOFA can be improved when measurements which attempt to determine other organ or systemic failures are added. To complement the interpretation and conclusions of our study, we have modified this section to warn readers that clinical applications could be limited at current time.

Was the parameters of the qSOFA score assessed as well?  This would be an interesting addition to this manuscript, as the qSOFA score is a fast measure, that doesn't require lab values, which might be more widely used in the EDs (although recent sepsis guidelines recommend not using as a screening tool) - I do see the qSOFA was commented upon in the discussion.

R: Thank you for commenting on this. Unfortunately, we did not contemplate to determine and assess qSOFA at the moment of designing our study. We agree that this could have been interesting and would provide important information, especially pertaining to clinical applications of prediction models. On a slightly different note, we thought that choosing a robust prediction scoring system like SOFA to compare the addition of Imp-R would be better than evaluating less robust scores since adding Imp-R to less robust scoring systems could have potentially made it easier to observe better predictions by adding one variable. Thus, we considered it necessary for us to evaluate Imp-R against a robust and comprehensive score. As you correctly mention, we had already commented this as a limitation of our study. We believe that by adding the explanations on the limited clinical applicability may better contextualize what our study adds while also considering its limitations.

Reviewer 3 Report

- In tables 2 and 3, the p-value should be corrected to use a standardized form of the result (3 decimal places). I also recommend bold text for statistically significant results

- the literature contains 33 items, but only 15 should be considered up-to-date (from the last 5 years). I recommend supplementing the literature with up-to-date results, e.g .:

a) Rahmatinejad Z, Reihani H, Tohidinezhad F, Rahmatinejad F, Peyravi S, Pourmand A, et al. Predictive performance of the SOFA and mSOFA scoring systems for predicting in-hospital mortality in the emergency department. Am J Emerg Med. 2019; 37 (7): 1237-1241.

b) Sosnowska-Mlak O, Curt N, Pinet-Peralta LM. Survival in sudden cardiac arrest in emergency room: case-control study. Crit. Care Innov. 2019; 2 (3): 1-10.

c) Krishna G, Kumar S, Sankar R, Raghu K, Sathynarayana V, Siripriya P. Sequential organ failure assessment and modified early warning score system versus quick SOFA score to predict the length of hospital stay in sepsis patients - accuracy scoring study. Crit. Care Innov. 2021; 4 (4): 9-18.

Author Response

April 21, 2022

Dr. Rahman Shiri

Editor-in-Chief

Healthcare

Thank you and the reviewers for kindly reviewing our paper entitled SOFA score plus impedance ratio predict mortality in critically ill patients admitted to the emergency department: Retrospective observational study. Manuscript ID: healthcare-1673018. We appreciate the thoughtful comments and suggestions which have allowed us to improve our manuscript. We have prepared detailed responses to all of your comments below.

Reviewer 3:

- In tables 2 and 3, the p-value should be corrected to use a standardized form of the result (3 decimal places). I also recommend bold text for statistically significant results

R: Thank you for this correction. We have standardized the p-value to decimal place and bold text for statistically significant results.

- the literature contains 33 items, but only 15 should be considered up-to-date (from the last 5 years). I recommend supplementing the literature with up-to-date results, e.g .:

  1. a) Rahmatinejad Z, Reihani H, Tohidinezhad F, Rahmatinejad F, Peyravi S, Pourmand A, et al. Predictive performance of the SOFA and mSOFA scoring systems for predicting in-hospital mortality in the emergency department. Am J Emerg Med. 2019; 37 (7): 1237-1241.
  2. b) Sosnowska-Mlak O, Curt N, Pinet-Peralta LM. Survival in sudden cardiac arrest in emergency room: case-control study. Crit. Care Innov. 2019; 2 (3): 1-10.
  3. c) Krishna G, Kumar S, Sankar R, Raghu K, Sathynarayana V, Siripriya P. Sequential organ failure assessment and modified early warning score system versus quick SOFA score to predict the length of hospital stay in sepsis patients - accuracy scoring study. Crit. Care Innov. 2021; 4 (4): 9-18.

R: Thank you for contributing with this suggestion. We have reviewed the suggested references and decided to add them to the manuscript since these are indeed relevant for the topic. Currently, our manuscript includes 36 references, 20 of which have been published within the last 5 years, and 11 of which have been published between the last 5-10 years. The remaining references older than 10 years are of much relevance to our manuscript, reason why we have decided to keep them.  

Reviewer 4 Report

Dear Authors, I woud like to thank you for this interesting paper.

It's simple in its design but clinically relevant for its implications.

I have only few questions for you, dictated by personal curiosity:

-how long is the learning curve to use the device?

-how long does it take the measurement of Imp-R with BODYSTAT? I ask you this because it's should be pivotal in ED where you need a rapid evaluation of the patient, while in ICU a time-consuming procedure is allowed.

Could you insert brief sentences in the paper to better specify these aspects pertaining the use of BODYSTAT?

Author Response

April 21, 2022

Dr. Rahman Shiri

Editor-in-Chief

Healthcare

Thank you and the reviewers for kindly reviewing our paper entitled SOFA score plus impedance ratio predict mortality in critically ill patients admitted to the emergency department: Retrospective observational study. Manuscript ID: healthcare-1673018. We appreciate the thoughtful comments and suggestions which have allowed us to improve our manuscript. We have prepared detailed responses to all of your comments below.

Reviewer 4:

Dear Authors, I woud like to thank you for this interesting paper.

It's simple in its design but clinically relevant for its implications.

I have only few questions for you, dictated by personal curiosity:

-how long is the learning curve to use the device?

R: Thank you for your comments. The learning curve for being able to use the impedance analyzer is approximately one week.

-how long does it take the measurement of Imp-R with BODYSTAT? I ask you this because it's should be pivotal in ED where you need a rapid evaluation of the patient, while in ICU a time-consuming procedure is allowed.

R: Thank you for your interest on this point. The complete evaluation takes approximately 5 minutes.

Could you insert brief sentences in the paper to better specify these aspects pertaining the use of BODYSTAT?

R: Sure, we have added a paragraph in the manuscript specifying the aspects pertaining the use of BODYSTAT. Thank you for suggesting this.

Round 2

Reviewer 2 Report

Thank you for your edits.